

# CFD Modeling of Reactive Pollutants Dispersion in Simplified Urban Configurations with Different Chemical Mechanisms

Beatriz Sanchez[1], Jose-Luis Santiago[1], Alberto Martilli[1], Magdalena Palacios[1], and Frank Kirchner[2]

[1]Research Center for Energy, Environment and Technology (CIEMAT), Madrid, Spain
[2]GAIASENS Technologies. Sarl, Switzerland

*Correspondence to:* Beatriz Sanchez (beatriz.sanchez@ciemat.es)

**Abstract.** An accurate understanding of urban air quality requires considering a coupled behavior between dispersion of reactive pollutants and atmospheric dynamics. Currently, urban air pollution is mostly dominated by traffic emissions and the primary emitted pollutants are nitrogen oxides ($NO_x$) and Volatile Organic Compounds (VOC). Modeling reactive pollutants with a large set of chemical reactions using a computational fluid dynamics (CFD) model requires a significant amount of CPU

time. In this sense, the selection of the chemical reactions needed in different conditions that gives the best compromise between CPU time and accuracy becomes essential. Three chemical approaches are considered: a) passive tracers (non-reactive), b) the $NO_x - O_3$ photostationary state, and c) a more complex chemical mechanism based on the RACM ('Regional Atmospheric Chemistry Mechanism') and it is reduced to 23 species and 25 reactions using CHEMATA software (Kirchner, 2005). The appraisal of the effects of the chemical reactions is focused on the study of NO and $NO_2$ dispersion and the comparison with

the tracer behavior within the street. Taking into consideration the VOC reactions, various $VOC/NO_x$ emission ratios of traffic are studied. In addition, the concentration of reactive pollutants is affected by many atmospheric parameters. In this work, the effect of the amount of background $O_3$ concentration depending on the season and different wind speeds are studied. Results show that the presence of ozone in the street acquires an important role in NO and $NO_2$ dispersion. Thus, greater differences linked to the chemical approach used are founded with higher $O_3$ concentration and faster wind speed. This is also related with

the vertical flux as a function of ambient wind speed since it improves the pollutants exchange among the street and overlying air. The joint evaluation of both parameters allows to ascertain the atmospheric conditions as from which the importance of the chemical reactions on NO and $NO_2$ concentration is significant. This is a detailed study that aims to understand the behavior of NO and $NO_2$ as reactive pollutants in several atmospheric conditions. The conclusions can be applied to future researches in order to determine the chemical reactions needed in terms of accuracy in modeling NO and $NO_2$ dispersion and the CPU

time required in a real urban area.

## 1 Introduction

Urban air pollution is nowadays a serious environmental problem. The non-uniformity in the distribution of buildings in a city involves complex flow patterns and therein, heterogeneous pollutant dispersion within the streets. In addition, the high levels of detrimental pollutants in urban areas are mostly dominated by traffic emissions. The main related-traffic pollutants are $NO_x$,



CO, hydrocarbons and particles. Due to the proximity between sources and receptors in the street, only the fastest chemical reactions have an impact on pollutants concentration. Therefore, some slow reactive compounds like CO or hydrocarbons can be considered as practically inert species at microscale. However, the dissociation of $NO_2$ in presence of light and the interaction between NO and $O_3$ take place rather fast (Vardoulakis et al., 2003). Besides that Volatile Organic Compounds

(VOC) are also involved in this complex chain of reactions.

Currently, modeling urban air quality using a computational fluid dynamics (CFD) model represents a big challenge. In addition to dispersion, the simulation of chemical transformations of reactive pollutants in a complex urban area increases considerably the computational time. To gain a better understanding of the problem, firstly we need to study thoroughly the basic dynamical effects controlling non-reactive pollutants dispersion in simplified geometries. Many previous studies have

investigated the main factors that affect the pollutants distribution within the canopy such as the inflow conditions (Tominaga and Stathopoulos, 2010; Kim and Baik, 2004), the street-canyon aspect ratio (Chang and Meroney, 2003), thermal effects of a heating surface (Park et al., 2012) or the vegetation location in the street (Buccolieri et al., 2011).

On the other hand, the relative importance of chemistry versus turbulent transport have been investigated in the past also for homogeneous surface layers (Galmarini et al., 1997; Molemaker and Vilà-Guerau de Arellano, 1998). The relative differences

between turbulent and chemical time scales were evaluated using the Damkholer number. However, we estimate that this approach is not appropriate in the urban canopy layer, basically because the transport is not only turbulent (as it is in the homogeneous surface layer), but also driven by the mean motions induced by the presence of the obstacles. Instead of adding a new time scale - linked to the mean transport - and try to deal with the strong heterogeneity of the flow (and associated time scales) typical of the UCL, we found more straightforward to perform simulations with and without the chemical reactions,

and assess the relative importance of chemistry based on the difference between the simulations.

The traffic-related pollutants are mostly reactive and for that reason, the photostationary scheme was included in an attempt to examine the dispersion of pollutants in street canyons. Baker et al. (2004) and Baik et al. (2007) studied the dispersion of reactive pollutants in a street canyon considering the $NO_x$-$O_3$ photostationary steady state. Baik et al. (2007) also incorporated a heating surface at the bottom of the street and found that the magnitude of the chemical term for $O_3$ was comparable to the

advection or turbulent diffusion terms, but that was not the case either for NO or for $NO_2$. Both researches showed that the $O_3$ concentration was faster depleted within the canopy because of the high NO emission at ground level. These studies underlined the importance of including the $O_3$ photochemical reaction in order to analyze the NO and $NO_2$ dispersion.

It has been shown that the primary pollutants emitted from vehicles are $NO_x$ ($NO + NO_2$) and VOC. For that reason, to model air pollution in an urban area becomes necessary to incorporate a complex chemical scheme that enables to define a

vast number of chemical interactions. In more recent studies, complex chemical mechanisms have been implemented in CFD models in order to reproduce the NO and $NO_2$ dispersion in a street canyon. Kwak and Baik (2012) developed a CFD model coupled with the carbon bond mechanism IV (CBM-IV) and evaluated the $O_3$ sensitivity with respect to different VOC and $NO_x$ emission levels in a street canyon. They found that the NO conversion to $O_3$ was more marked than the $NO_2$ photolysis given by the continuous emission from vehicles, which causes that the $O_3$ behavior has a negative correlation with $NO_x$

emission level. Likewise, the $O_3$ sensitivity was weakly correlated with the amount of VOC emission. Kwak et al. (2013)





examined the dispersion and photochemical evolution of reactive pollutants in a street canyon with different canyon aspect ratios. It confirmed the $O_3$ relation with $NO_x$ and VOC emission levels and concluded a slight influence of wind speed whereby increasing the wind speed, the exchange with overlying air enhances and strengthen the downward $O_3$ transport. These recent studies highlight the importance of $O_3$ oxidation process as well as the OH oxidation process in describing the photochemical

reactions in the street. Park et al. (2015) analyzed different aspect ratios and several VOC/$NO_x$ emission scenarios in three-dimensional street canyons. The main effects of $NO_x$ and VOC emission levels in $O_3$ concentration are similar to the results obtained in two-dimensional studies. In contrast, small variations are locally found owing to different flow patterns generated by the geometry of the domain.

To further analyze the sensitivity of the model with the chemical mechanism, Kim et al. (2012) compared the hourly concen-

tration of NO, $NO_2$ and $O_3$ with the photostationary steady state and a full chemical mechanism (110 species and 343 reactions) with identical conditions. The results revealed similar values of NO and $NO_2$ concentration, while the $O_3$ concentration with the simple mechanism was lower than that obtained with the full scheme. Bright et al. (2013) compared the $O_3$-$NO_x$ chemical system with a full chemical mechanism (51 chemical species and 136 reactions) using a LES model. They studied in depth the effects of VOC chemical processes and found that by means of a more detailed chemistry, the values of NO, $NO_2$ and

$O_3$ differ from the photostationary steady state, which reflects additional transformation of NO into $NO_2$ resulting from the intrinsic VOC degradation processes.

Despite the fact that a full chemical scheme involves a large number of reactive species and chemical reactions that can enhance the description of the chemical processes in the atmosphere, modeling the reactive pollutant dispersion in a real urban area increases the computational time considerably, even more than a factor 2 with respect to a simple chemical scheme.

Addressing this goal first requires to establish some criteria to define when the differences in pollutant concentration among chemical schemes are insignificant. Considering the impact of external conditions on the concentration of reactive pollutants, the effects of the amount of $O_3$ of background depending on the season, the wind speed and the VOC/$NO_x$ emission ratio are jointly assessed. The aim of this work is to determine in which conditions result essential to incorporate more chemical reactions in order to model the dispersion of reactive compounds in the urban canopy. This goal is reached by analyzing the

differences in NO and $NO_2$ concentration obtained with the photostationary steady state and with a more complex chemical mechanism in two and three dimensional simplified geometries using a CFD-RANS model.

## 2   Chemical Schemes Used

In this work, the variation on NO and $NO_2$ concentration linked with the use of the photostationary steady state or a complex chemical mechanism is quantified in comparison to NO and $NO_2$ regarded as non-reactive pollutants. The impact of including

the $O_3 - NO_x$ reactions is assessed by comparing the deviation with a passive tracer, whereas the importance of including reactions involving VOC is derived by evaluating the difference with respect to the $NO_x - O_3$ system. The photostationary steady state is the most simple chemical mechanism implemented in the CFD and it is defined by a three-reaction system of





NO, $NO_2$ and $O_3$.

$$NO_2 + h\nu \longrightarrow NO + O \tag{R1}$$

$$O + O_2 + M \longrightarrow O_3 + M \tag{R2}$$

$$O_3 + NO \longrightarrow NO_2 + O_2 \tag{R3}$$

Where M represents a molecule that absorbs excess energy and thereby stabilizes $O_3$ molecule formed (Seinfeld and Pandis, 1998). The photolysis rate ($J_{NO_2}$) and reaction rate constant ($k$) are dependent on zenith angle ($\theta$) and temperature respectively.

On the other hand, a complex chemical mechanism for the CFD model (CCM-CFD) is developed. Due to the limitation of
10 CPU time, the RACM have been reduced to 25 reactions for 23 chemical species in order to reproduce the composition of the urban atmosphere with the CFD model. The complex chemical mechanism (CCM-CFD) was created applying the software CHEMATA (CHEmical Mechanism Adaptation to Tropospheric Applications)(Junier et al., 2005; Kirchner, 2005). The aim was to reduce the RACM mechanism as far as possible without losing much accuracy during the first 30 minutes of simulation for urban conditions in a boxmodel. A wide set of different urban pollution conditions was investigated and special emphasis
15 was given to cases where the results for the RACM mechanism differ considerable from the results of the photostationary steady state. Three main manners of mechanism reduction are applied:

 – Modification of the lumping groups: Several chemical similar RACM lumping groups are united to a new lumping group. For example, the RACM alkanes ETH, HC3, HC5 and HC8 are summed up in the new species ALK. The RACM olefins ETE, OLT, OLI are lumped to the new species OLE and the RACM aromatic compounds TOL and XYL are integrated
20   in the new species ARO.

 – Modification of the organic peroxy radical parametrisation: instead of using the 20 RACM RO2 species, we applied the approach described by Kirchner (2005) with the RO2 species CH3O2 (called MO2) and RC(O)O2 (called ACO3) as well as the two operator species XO2 and XO2N which work in the following way. The real reactions:

$$R - H + HO \longrightarrow R^* + H_2O \tag{R4}$$

$$R^* + O_2 \longrightarrow RO_2 \tag{R5}$$

$$RO_2 + NO \longrightarrow a\, RONO_2 + b\,(RO + NO_2) \tag{R6}$$





$$RO \longrightarrow Products \tag{R7}$$

are parameterised by;

$$R-H + HO \longrightarrow a\,XO_2N + b\,(XO_2 + products) \tag{R8}$$

$$XO_2 + NO \longrightarrow NO_2 \tag{R9}$$

$$XO_2N + NO \longrightarrow RONO_2 \tag{R10}$$

 – Elimination of reactions and species which are not crucial for simulating under the given conditions (time scale of about
30 minutes, urban pollution conditions, daytime). Focusing on daytime simulations we can eliminate the species $NO_3$
 and all its reactions. As well low reactive species as $CH_4$ and $H_2$ can be eliminated under urban conditions.

The resulting CCM mechanism is presented in Table 1. Compared to the photostationary steady state this CCM-CFD mechanism includes the peroxy radical production from VOC, CO and $SO_2$ as well as $NO_x$ loss processes by the formation of PAN, $HNO_3$ and organic nitrates. Whereas the peroxy radical production increases the $NO_2/NO$ ratio as therefore the $NO_2$
concentrations, the $NO_x$ loss processes decrease $NO_x$ and therefore $NO_2$. As well, $O_3$ loss by the reaction with olefins is considered.

## 3  CFD Model Description

### 3.1  Model Description

The CFD model used is based on the Reynolds-averaged Navier-Stokes equations (RANS) with a k-$\varepsilon$ turbulence model. This
CFD-RANS model is STARCCM+ (from CD-ADAPCO). It has been used in previous researches in order to study dynamical processes, thermal forcing and passive tracers dispersion in simplified and real urban geometries (Santiago and Martilli, 2010; Martilli et al., 2013; Santiago et al., 2014). The $NO_x$-$O_3$ photostationary state and the complex chemical mechanism described in previous section are implemented in the CFD model in order to simulate and evaluate the pollutants dispersion incorporating different chemical approaches.
In this work, the transport equations are modified for reactive chemical species. The chemical term is defined as formation and depletion of a compound in the chemical reactions. It is introduced as a production rate in the transport equation for each chemical specie ($[\Delta C]_{Chem}$) in the CFD model. The traffic emission is added like a source term ($S_{C_i}$). Hence the transport



equations are given by,

$$\frac{\partial C_i}{\partial t} + U_i \frac{\partial C_i}{\partial x_j} = D \frac{\partial^2 C_i}{\partial x_j \partial x_j} + \frac{\partial}{\partial x_j}\left(K_c \frac{\partial C_i}{\partial x_j}\right) + \left[\Delta C\right]_{Chem} + S_{C_i} \tag{1}$$

Where $C_i$ is the concentration of the ith specie and $D$ and $K_c$ are molecular diffusivity and eddy diffusivity of pollutants, respectively.

## 3.2 Simulation Setup

The distribution of pollutant and the residence time of the pollutant in the street is mainly affected by the flow defined as a result of buildings configuration. The pollutant dispersion is analyzed in two types of geometries with the purpose of understanding the behavior of reactive pollutants inside the street and the dependency with the geometry. Figure 1 shows the computational domains used: a single street-canyon (2D-geometry) and a staggered array of cubes (3D-geometry). The street-canyon and staggered array of cubes domain sizes are 24x40x64 m and 64x64x64 m in x, y and z directions respectively. The aspect ratio defined as the ratio of building height ($H$) to the street width ($W$) is $H/W = 1$ in both geometries. The top of the domains is located at $4H$ and the grid resolution is 1 m in all directions. The three-dimensional array of cubes is represented by staggered cubes with buildings width ($L$), resulting a packing density $\lambda = 0.25$.

Symmetry boundary conditions are assumed in the spanwise direction (y-direction) and cyclic boundary conditions are imposed in the streamwise direction in order to simulate an infinite number of streets. The flow is driven in x-direction by pressure gradient equal to $\rho u_\tau^2/4H$, where $u_\tau$ is a reference velocity. For this work, two velocities are studied, $u_\tau$=0.45 m s$^{-1}$ and $u_\tau$=0.23 m s$^{-1}$. At the top, symmetry conditions are established that enforce parallel flow and zero normal derivatives for the dynamic variables are imposed (Santiago and Martilli, 2010).

For each geometry, the pollutants dispersion is analyzed considering different chemical approaches: passive tracer, NO$_x$-O$_3$ photostationary state and NO$_x$-O$_x$-VOC chemical reactions with VOC/NO$_x$ emission ratio equal to 1/5 and 1/2. These values are within the typical values of emission ratio in the last years for the traffic of Madrid city (Ayuntamiento-Madrid, 2014). These four chemical scenarios are simulated under different atmospheric conditions, considering two values of wind speeds and two conditions of O$_3$ concentration of background (Table 2).

Emission sources are located at ground level in both domains (Fig.1). Traffic emissions are simulated considering a NO$_x$ emission of 0.5 g km$^{-1}$ per vehicle (Baker et al., 2004; Baik et al., 2007) and the volumetric ratio of NO and NO$_2$ emission is 10:1 (Buckingham, 1997). Thus NO and NO$_2$ emissions are fixed with rates of 112 μg m$^{-1}$s$^{-1}$ and 17 μg m$^{-1}$s$^{-1}$ respectively and it is equivalent to 930 vehicles per hour representative of medium traffic. In addition, the complex chemical mechanism requires to consider VOC emissions. The ratio VOC/NO$_x$ emitted by vehicles is dependent on several factors such as kind of vehicle or traffic flow speed. For that, two VOC/NO$_x$ emission ratios, 1/5 and 1/2, are simulated. Some VOC are lumped in terms of specific functional groups in the complex chemical scheme. The emitted VOC species considered are OLE, ARO, ALK, ALD and HCHO, and their volumetric proportions are 28.6 %, 23.1 %, 38.6 %, 4.0 % and 5.6 %.

Due to periodic and symmetry conditions along x-direction and y-direction respectively, the concentrations at the top play an important role considering chemical reactions. The background concentrations of reactive species are imposed at top of the





domain. The values of NO, $NO_2$, CO and $SO_2$ concentration are fixed at 16, 35, 200 and 2 ppb for all scenarios. However, VOC concentrations are established on the same proportion of $VOC/NO_x$ emission ratios.

Overall, the diurnal variation of traffic flow in an urban area register a peak of traffic at 08:00-09:00, producing a valley at 13:00-14:00 and increasing gradually to another peak value at 20:00-21:00. As a consequence, the NO and $NO_2$ concentrations (hereafter, referred as [NO] and [$NO_2$]) have the same time evolution. For that reason, we focused on analyzing the NO and $NO_2$ dispersion in maximum conditions of traffic emissions. In contrast, the diurnal evolution of $O_3$ concentration (hereafter, referred as [$O_3$]) inside the street follows the opposite behavior owing to the photochemical interactions. In turn, the amount of $O_3$ of background is dependent on the solar radiation. With the purpose of evaluating the influence of the available $O_3$ on [NO] and [$NO_2$] within the street, two characteristic cases of opposite seasons have been studied. For that, a representative zenith angle of winter, 78°, and another one of summer, 46°, are considered in order to compute the [$O_3$] established at the top of the domain at 08:00 LST using the photochemical equilibrium equation,

$$[O_3] = \frac{J_{NO_2}[NO_2]}{k[NO]} \tag{2}$$

Where [NO] and [$NO_2$] are constant at the top in all cases. The assumption of isothermal condition is applied considering the air temperature equal to 298 K, so the rate reaction constant ($k$) is set at all simulations. However, the photolysis rate $J_{NO_2}$ is calculated as a function of the zenith angle ($J_{NO_2} = A\,exp(B/cos(\theta))$, where A and B are chemical constants). The selected angles representative of winter and summer are 78° and 46° resulting in 10.6 and 39.8 ppb of [$O_3$] of background respectively.

### 3.3 CFD Model Evaluation

Due to a lack of experimental data that can be used to evaluate the chemical mechanisms, the results from the CFD model are evaluated in two steps. Firstly, the dispersion in the canyon using the non-reactive case is compared with experimental data (Meroney et al., 1996). Secondly, the implementation in the CFD model of the chemical terms as a part of the complex chemical scheme (CCM-CFD) is evaluated with the results from a chemical box model well-established (Junier et al., 2005; Kirchner, 2005).

Despite the fact that the non-reactive pollutant dispersion was previously validated with this CFD model (Santiago and Martín, 2008; Martilli et al., 2013), the current results are also evaluated against wind tunnel experimental data (Meroney et al., 1996). The experimental system consists on a single 2D street canyon with aspect ratio equal to unity. The wind blows perpendicular to the canyon in x-direction and the emission source is located at the bottom of the domain. This scenario is similar to the 2D-CFD simulations with the exception of the cyclic boundary conditions imposed in streamwise direction that simulate an infinite number of streets with their corresponding emissions. This is taken into account in the comparison with experimental data by dismissing the inlet concentration from the concentration inside the canyon.

We focus on comparing pollutant concentration from simulation results with the measurements of urban roughness experimental case at different points close to walls within the canyon (Fig. 2). The concentration at each measurement point is normalized with respect to the concentration at the point 7 within the street ($C/C_7$). The same normalization was used in





Santiago and Martín (2008) to evaluate the CFD model results with same experiment. Comparison between experimental and computed normalized concentration is depicted in Fig. 3.

Overall, a good agreement is obtained in spite of the slight difference at point 4 and 13 where the concentration is lower than experimental data. Besides the linear regression concludes a good correlation (R=0.92) between experimental measurements and simulated results. For further information, the statistical parameters of Normalized Mean Square Error (*NMSE*), Fractional BIAS (*FB*), correlation (*R*) and the fraction of predictions within a factor of 2 of observations (*FAC2*) are computed following the equations,

$$NMSE = \frac{\sum_{i=1}^{n}(O_i - P_i)^2}{\sum_{i=1}^{n}(O_i P_i)} \tag{3}$$

$$FB = \frac{\overline{O} - \overline{P}}{0.5(\overline{O} + \overline{P})} \tag{4}$$

$$R = \frac{\sum_{i=1}^{n}[(O_i - \overline{O})(P_i - \overline{P})]}{\left[\sum_{i=1}^{n}(O_i - \overline{O})^2\right]^{1/2}\left[\sum_{i=1}^{n}(P_i - \overline{P})^2\right]^{1/2}} \tag{5}$$

$$FAC2 = \text{fraction of data that satisfy } 0.5 \leq P_i/O_i \geq 2 \tag{6}$$

Where $n$ is the number of points, the $O_i$ are the measurements at each point, and $\overline{O}$ is the measurement mean. And $P_i$ and $\overline{P}$ are the computed values at each point and its corresponding mean. The values of *FAC2*=1, *NMSE*=0.030 and *FB*= 0.068 reveal a good fit between computed and experimental data with a slight underestimation. Therefore, the pollutant dispersion is accurately simulated in base to the model acceptance criteria established for *NMSE*<1.5 and -0.3<*FB*<0.3 in Chang and Hanna (2005) and as a threshold of the correlation coefficient *R*>0.8 in Goricsán et al. (2011) (Table 3).

Secondly, the accuracy of solving chemical reactions in the CFD model is assessed in comparison with the simulation results using a chemical box model described in Sect. 2. The chemical evolution for all reactive species is modeled during 30 min considering the initial conditions and the related-traffic pollutant emissions of an urban area. In both cases, the same reaction rate and the photolysis rate constants are used considering the temperature of 293 K and a constant zenith angle of 40°. In a detail assessment, the statistical parameters *NMSE*, *FB*, *R* and the maximum relative concentration error from the CFD results with respect to the box model results are calculated for all pollutants, but only for NO, $NO_2$ and $O_3$ are shown in Table 3. The *NMSE* and *FB* indicate that the values of concentration are quite similar, besides of the correlation is almost 1 for all pollutants. As well as the maximum difference of relative error of the concentration of all chemical species is less than 3 %. These differences are caused mainly for numerical errors. Therefore, the implementation of the chemical terms in the transport equations in the CFD model seems to be appropriately solved.

Lastly, the appropriate time step in order to solve the dynamic transport and the chemical reactions is evaluated using the complex chemical mechanism (CCM-CFD (VOC/$NO_x$=1/2) in the street canyon geometry. The concentration of reactive





pollutants is analyzed for different time steps (0.1 s, 1 s and 2 s). During the first period of the simulation, the turbulent dynamic is simulated with 1 s of time step, without taking into account chemical interactions. After that, the chemical reactions are implemented in the CFD simulation carrying out every time step case ($t_s$=0.1 s, $t_s$=1 s and $t_s$=2 s). Finally, when the steady state is reached, the horizontal spatial average concentrations of NO, $NO_2$ and $O_3$ are compared in the whole domain. The deviation of concentration from the case with $t_s$=1 s is quantified using the quadratic relative differences given by,

$$QD(\%) = \frac{\sqrt{\frac{1}{n} \sum_{k}^{n} (C'_k - C_k)^2}}{\frac{1}{n} \sum_{k}^{n} C_k} \cdot 100 \qquad (7)$$

where $C'_k$ and $C_k$ is the horizontal average concentration at several vertical levels with $t_s$=0.1 s or 2 s and $t_s$=1 s, respectively. The results of the $QD$ for NO, $NO_2$ and $O_3$ from the simulations with the time step 0.1 s and 2 s are enclosed in Table 5. By comparing these results, we establish the time step at 1 s in order to simulate the dispersion of reactive pollutants.

## 4   Results and discussion

The CFD model is used to simulate turbulent flow and dispersion of reactive species in two different geometries: a 2D street canyon and a 3D staggered array of cubes. The interaction between the atmosphere and buildings configuration induces complex flow patterns within the urban canopy and this induces a heterogeneous pollutant distribution inside the streets. Apparently, in a 2D-geometry the ventilation of the street is less than in a 3D-geometry, and the pollutants are maintained more time within the street. Therefore, the residence time of each reactive compound within street is determined by building configurations and wind speed. The time that these pollutants stay inside the street affects to the chemical interactions, and as a consequence, to the amount of concentration at pedestrian level. With the objective to deepen on this dependency with the geometry, the pollutant concentration in both domains are compared. To facility the comparison, the concentration of the study pollutants are normalized using the emission area ($A_{Em}$), the reference velocity ($u_\tau$) and the source emission rate ($Q$) given by,

$$C_{norm} = \frac{C\, u_\tau\, A_{Em}}{Q/L} \qquad (8)$$

This work is focused on analyzing NO and $NO_2$ dispersion within the street and their normalized concentration are referred as $[NO]_N$ and $[NO_2]_N$. In addition to quantify the effect of considering chemical reactions, the deviation in concentration ($\delta C$) from the non-reactive compound behavior is computed as,

$$\delta C(\%) = \frac{C_{norm} - C_{norm}(T)}{C_{norm}(T)} \cdot 100 \qquad (9)$$

where $C_{norm}$ and $C_{norm}(T)$ are the normalized concentration of the pollutants regarded as reactive and non-reactive, respectively.





### 4.1 Ozone Influence on Reactive Pollutants

The $O_3$ concentration is generally dependent on the photochemical regime but in the street, it is also affected by chemical interactions with the $NO_x$ and VOC levels owing to the VOC-$NO_x$ traffic emission ratio. The effect of various VOC-$NO_x$ emission ratios setting the amount of background $[O_3]$ was analyzed in Kwak and Baik (2012). In addition, with a $NO_x$-VOC

emission ratio fixed, the diurnal variation of $NO_x$ and the exchange of $O_3$ at roof level was studied in Kwak and Baik (2014). To further examine the influence of the available $O_3$ on [NO] and $[NO_2]$ in the street with several levels of VOC emission, two zenith angles representative of winter ($78°$) and summer ($46°$) are used in order to model two background concentrations of $O_3$, 10.6 and 39.8 ppb respectively (Table 2).

Figure 4 shows $\delta NO$ and $\delta NO_2$ in the 2D geometry. In the case with lower $O_3$ of background, either considering the

photostationary steady state (PSS) or the CCM-CFD with VOC/$NO_x$ emission ratio 1/5 or 1/2 (hereafter referred as CCM1/5 and CCM1/2), $\delta NO$ and $\delta NO_2$ below the canopy are around 3 % and 30 % respectively. This indicates that the reactions involving VOC have little impact on NO and $NO_2$ levels, whereas the $NO_x - O_3$ reactions have a major effect on $[NO_2]$. On the other hand, in the higher $O_3$ case, $\delta NO$ and $\delta NO_2$ increase in average to nearly 12.5 % and more than 100 % inside the street, and besides the differences between chemical scenarios are remarkable. As a result, in the higher $O_3$ case and high VOC

emission, the $[NO_2]$ and [NO] in the street is respectively underestimated and overestimated with the photostationary steady state in comparison with the results including the VOC reactions.

To evaluate the variation of adding VOC reactions instead of just considering the $NO_x - O_3$ system with the CFD model, the $[NO]_N$ and $[NO_2]_N$ from PSS, CCM1/5 and CCM1/2 are compared in normalized concentration with their corresponding non-reactive pollutant below the canopy in the 2D and 3D geometries. Figure 5 illustrates the vertical profiles of the horizontal

spatial average of $[NO]_N$ and $[NO_2]_N$ for the two $O_3$ cases. In the winter case, the difference between chemical scenarios is negligible. Overall, the variation of $[NO]_N$ and $[NO_2]_N$ among PSS, CCM1/5 and CCM1/2 are bounded on 0.05 % and 2.3 %, which represents a maximum deviation of 0.3 and 1.3 ppb in absolute concentration within the street respectively. In contrast, in the higher $O_3$ case, the importance of the photochemical reactions increases. With the PSS, CCM1/5 and CCM1/2, the $[NO]_N$ is respectively about 10.1 %, 10.5 % and 12.3 % less than the tracer normalized concentration within the canopy

in the street canyon geometry. Likewise in the 3D-geometry, the $[NO]_N$ for all mechanisms are shifted around 10-12 % from non-reactive pollutant. Focusing on $[NO_2]_N$ including VOC reactions against the $NO_x - O_3$ system, $NO_2$ is more affected by the increase of background $O_3$ and the photochemical reactions owing to the zenith angle of summer. The $[NO_2]_N$ rises more than 15 % with CCM1/2 which represents values of up to 10 ppb in the 2D configuration and 7 ppb in the staggered array of cubes. Therefore, the $[O_3]$ of background and the solar position ($\theta$) regarded in photolysis rates highlight the effect of including

more chemical reactions on the study of NO and $NO_2$ dispersion in the street.

In addition, the difference among chemical mechanisms is also influenced by VOC levels. In the lower $O_3$ case, the effect of VOC emissions have a minimum impact on $[NO]_N$ and $[NO_2]_N$. Hence the photostationary state or a more complex chemical mechanism can be used to reproduce the NO and $NO_2$ dispersion in the street since the results are quite similar. However, in the summer case with high VOC levels, the reactions involving VOC are more important and larger differences in [NO] and $[NO_2]$





with respect to $NO_x$-$O_3$ system are found. For instance in 2D-geometry, $[NO]_N/[NO(T)]_N$ is 0.87, 0.89 and 0.90 whereas $[NO_2]_N/[NO_2(T)]_N$ varies from 2.18, 2.03 and 2.01 for CCM1/2, CCM1/5 and PSS respectively. Concluding that taking into account the VOC reactions, the NO production is reduced and $NO_2$ retrieval is increased in comparison to the results from the PSS owing to the degradation of VOC. This effect is remarkable in the higher $[O_3]$ case and high VOC traffic emissions.

## 4.2 Wind speed influence on reactive pollutants

The concentration of a non-reactive pollutant is inversely proportional to wind speed. However, the non-linearity of chemical reactions introduces changes in the concentration behavior. To study the effect produced by wind speed on NO and $NO_2$ dispersion, the cases with different wind speeds are compared (Table 2). As discussed in the previous section, the higher $O_3$ case is used in this section given that chemical reactions have more effect on NO and $NO_2$ concentrations below the canopy.

Figure 6 shows the $\delta NO$ and $\delta NO_2$ for different values of wind speed in the street canyon geometry. With the faster velocity, the differences on normalized concentration are larger in comparison to the tracer behavior. That is because a major amount of $O_3$ is introduced into the street acting as a trigger of the chemical reactions between pollutants. As a conclusion in terms of normalized concentration, a major effect on NO and $NO_2$ is observed in comparison with a non-reactive specie with the higher velocity. Overall for all chemical scenarios, $[NO]_N$ is shifted around 18 % and 10 % respect to tracer with $u_\tau$=0.45 m s$^{-1}$ and $u_\tau$=0.23 m s$^{-1}$ respectively in the street. Whereas $[NO_2]_N$ is increased a factor 2.7 and 2 for $u_\tau$=0.45 m s$^{-1}$ and $u_\tau$=0.23 m s$^{-1}$. Therefore, a greater influence of chemical reactions on NO and $NO_2$ is obtained with the faster wind speed. Note that for a tracer, the normalized concentration is proportional to concentration and inversely proportional to wind speed. However, the effect of chemical reactions is not proportional to wind speed.

To analyze the concentrations of NO and $NO_2$ below the canopy given by the chemical reactions, the vertical profiles of horizontal spatial average of the normalized concentration are depicted in Fig. 7. The effects of chemical reactions with different wind speed are quite similar in 2D and 3D geometries. In the faster wind speed case, the normalized concentration tend to be further from the non-reactive normalized concentration, however the variation among chemical scenarios is small. In contrast with lower wind speed, the accumulation of $NO_x$ emissions is greater within the street which contributes to quickly consume the $[O_3]$, reducing the $NO_2$ production and the NO removal. Likewise, the VOC levels given by the increase of emissions in the street intensify the $NO_2$ formation and NO depletion, producing the opposite effect. Therefore, in the $u_\tau$=0.45 m s$^{-1}$ case, the results of $[NO]_N$ and $[NO_2]_N$ in regards to the non-reactive compound are increased around 20 % for $[NO]_N$ and more than double of $[NO_2]_N$, but between chemical systems differences are around 2 %. However, the largest differences between the chemical scenarios are observed with the lower wind speed. This means that the effect of considering reactive pollutants is enlarged with a faster wind speed because of a greater exchange of pollutants in the street is produced. Whereas when the wind speed is reduced, the importance of the reactions involving VOC increases in comparison with just regarding the $NO_x - O_3$ system.





### 4.3 Vertical transport

The vertical transport of NO, $NO_2$ and $O_3$ is analyzed in both geometries in order to determine the atmospheric conditions as from which the use of VOC reactions is needed to reproduce on accurate way the NO and $NO_2$ dispersion within the street. With this analysis, it is possible to understand how either wind speed or the background of $[O_3]$ affect on pollutants concentration. Figure 8 shows the average of the deviation of $[NO]_N$ and $[NO_2]_N$ below the canopy in comparison to their corresponding tracer in each geometry (Eq.9).

In the lower $O_3$ case, the differences on $\delta NO$ and $\delta NO_2$ among chemical mechanisms in 2D-geometry are negligible even with different wind speeds. From all chemical scenarios, the deviation with the tracer is around 5 % and 3 % for $u_\tau$=0.45 m s$^{-1}$ and $u_\tau$=0.23 m s$^{-1}$, respectively. Without hardly differences, the same behavior of NO and $NO_2$ is obtained in 3D geometry. However, in the higher $O_3$ conditions, the differences of the simple scheme and by including VOC reactions are greater and the dependency with the wind speed is significant. As it has seen before, despite of the deviation in comparison with tracer is larger with the $u_\tau$=0.45 m s$^{-1}$, the higher differences between chemical scenarios are obtained with $u_\tau$=0.23 m s$^{-1}$. The maximum differences on $[NO]_N$ and $[NO_2]_N$ are obtained with more VOC emissions, and their variation switch from 10 % to 20 % when the inflow velocity changes of $u_\tau$=0.45 m s$^{-1}$ to the half. Therefore, the chemical reactions are strongly determined by the available $O_3$ of background and subsequently, they have an slight dependency with the wind speed.

With the objective to deepen on the exchange of ozone inflow into the street, the vertical fluxes are analyzed. Given that, there are only small differences between both computational domains with the same aspect ratio H/W=1. The vertical transport is evaluated in the 3D-geometry. By means of vertical fluxes of NO, $NO_2$ and $O_3$ in all cases, the pollutant removal from the street and pollutant entrainment into the street are analyzed. The total vertical flux ($F_{T,i}$) is defined as a sum of vertical mean flux ($F_{m,i}$) and turbulent flux ($F_{t,i}$). The horizontal spatial average ($<>$) of $F_{T,i}$ of $i$th species is computed at several vertical levels every 1 m using the following equations.

$$F_{T,i} = <\overline{wC}> = F_{m,i} + F_{t,i} \tag{10}$$

$$F_{m,i} = <\overline{w}\,\overline{C_i}> \tag{11}$$

$$F_{t,i} = <\overline{w'C_i'}> = -K_c \frac{\partial C_i}{\partial z} \tag{12}$$

Here $w$ and $w'$ are the vertical velocity and its fluctuation respect to the mean, $\overline{w}$. And $C_i$ and $C_i'$ are the corresponding variables to the concentration of $i$th species. To examine the atmospheric parameters effect on NO and $NO_2$ regarded as reactive pollutants through vertical fluxes, the NO and $NO_2$ fluxes are normalized following the method developed for a passive tracer in Martilli et al. (2013). For a passive tracer, taking into account an homogeneous emission at surface, the steady state and neglecting vertical variations of air density with height, the following holds, within the canopy (Eq. 13) and above (Eq. 14).

$$<\overline{w'C'}> + <\overline{w}\,\overline{C}> = S \tag{13}$$





$$< \overline{w'C'} > + < \overline{w}\,\overline{C} > = S \frac{A_{street}}{A_{street} + A_{roof}} \tag{14}$$

Where S is the constant emission flux at the surface and $A_{street}$ and $A_{roof}$ are the areas of the streets and roofs of buildings respectively. Hereafter the spatial horizontal average of total fluxes of NO and $NO_2$ are normalized with S (within the canopy)

and S $A_{street}/(A_{street} + A_{roof})$ above, referred as $< \overline{w\,NO} >_N$ and $< \overline{w\,NO_2} >_N$.

Figure 9 illustrates the vertical profiles of the deviation of $< \overline{w\,NO} >_N$, $< \overline{w\,NO_2} >_N$ from the normalized total flux of tracer and the vertical profile of $< \overline{w\,O_3} >$ spatially averaged for the CCM1/2 and PSS scenarios in the 3D-geometry. The CCM1/5 results are not shown since their results are quite similar to PSS results. The $< \overline{w\,NO} >_N$ and $< \overline{w\,NO_2} >_N$ taking into account chemical reactions are compared with the normalized flux of tracer. The vertical transport of a tracer above the canopy is only

turbulent, given that the mean vertical velocity is zero, but within the canopy in the middle part, it is mainly dominated by mean flux. This behavior is obtained for NO and $NO_2$ regarded as reactive pollutants as well. Unlike the tracer fluxes, the turbulent fluxes of NO and $NO_2$ are affected by chemical interactions, mainly with $O_3$. In Fig. 9, we can observe that the effect of the chemical reactions involving VOC (with the VOC/$NO_x$=1/2) on vertical transports of NO and $NO_2$ are significant in the higher $[O_3]$ and faster wind speed case. In this case, the $< \overline{w\,NO} >_N$ and $< \overline{w\,NO_2} >_N$ are slightly minor and major

respectively, corresponding to a greater vertical transport of $O_3$ in all heights and more photochemical activity. In contrast, in the other cases, the influence of introducing chemical reactions on $< \overline{w\,NO} >_N$ and $< \overline{w\,NO_2} >_N$ just can be observed above the canopy whereas inside the street is quite similar to the tracer flux. That is due to fact that the $[O_3]$ is practically depleted by reaction with NO before entering into the street. Besides the strong NO emission and lower wind speed avoid the escape of NO from the street and block the $O_3$ formation by means of degradation reactions of VOC. That means that in the higher

$O_3$ case with the faster wind speed, the downward motion contributes to introduce $[O_3]$ into the street and the upward motion remove the [NO] of the street. And moreover, the VOC emissions benefit the conversion of NO to $NO_2$. Therefore, the impact of the reactions involving VOC with the emission ratio VOC/$NO_x$=1/2 on [NO] and $[NO_2]$ is significant in the street .

For a better understanding of the relative importance of including more chemical reactions and so to characterize the NO and $NO_2$ dispersion under several conditions, the effect on $[NO]_N$ and $[NO_2]_N$ within the canopy is analyzed considering jointly

all variables here studied. On this way, the variability of vertical transport of $O_3$ and in turn the effect of VOC emission, linked to the change of wind speed are examined. Figure 10.a shows the averaged concentration within the canopy of $[NO_2]_N$ over $[NO]_N$ in relation to the total flux of $O_3$ at roof level ($< \overline{w\,O_3} >_N$) for the PSS and CCM1/5 and CCM1/2 in the ozone and wind speed cases. Note that $[NO_2]_N/[NO]_N$ is 1 for the non-reactive species. Thus, the deviation from 1 indicates the effect of chemical reactions. Figure 10.b illustrates the variation of considering VOC reactions aside from the $NO_x$-$O_3$ reactions by

means of $[NO_2]_N$ with respect to the increase of VOC average concentration below the canopy.

In Fig. 10.a we can observe that the effect of chemical reactions is greater in the higher $O_3$. With faster wind speed, the inflow of $O_3$ to the street increases owing to the downward motion and for that, the value of $[NO_2]_N/[NO]_N$ is large. On the other hand, when the $[O_3]$ levels are not enough to reacts with large $NO_x$ emissions and the photochemical reactions are slower, the difference on $[NO]_N$ and $[NO_2]_N$ with respect to the normalized concentration tracer is smaller ($[NO_2]_N/[NO]_N < 1.75$).



Focusing on the lower $O_3$ case, the variation among chemical scenarios is negligible. However by increasing the inflow of $[O_3]$ into the street, higher differences are found on $[NO_2]_N/[NO]_N$. To deepen on the deviation of the use of VOC reactions against to just the photostationary steady state, the influence of the VOC average concentration below the canopy is analyzed in Fig. 10.b The slight variations among chemical scenarios is just caused by VOC levels. Although the results conclude that small differences exist in winter conditions, they are insignificant since the maximum deviation is 1 ppb corresponding to the slow velocity for CCM1/2. In contrast, in summer case, with high $[O_3]$, the variations between chemical scenarios are greater and increase with the VOC emission in the street. That is due to the slower wind speed which induces less vertical exchange and avoids the exit of VOC concentration from the street. In fact the higher VOC emissions under these conditions benefit the VOC reactions, which induce the conversion of NO to $NO_2$, rising the difference with respect to the $NO_x$-$O_3$ reactions.

In summary, the impact of the chemical reactions in the canopy is affected by the $O_3$ entrainment within the canopy. And this is directly proportional to the value of $O_3$ above the canopy and the wind speed. On the other hand, the VOC reactions are more important in the canopy under high $O_3$ and high VOC levels. The second depends on the emission ratio and the upward VOC flux at the top of the canopy. For that, stronger winds will increase the flux, and so reduce the importance of the VOC reactions in the canopy.

## 5 Summary and Conclusions

With the aim of optimizing the modeling of NO and $NO_2$ dispersion over a real urban area an analysis of the relative importance of chemical reactions respect to mean and turbulent transport has been carried out by means of a CFD model. This is based on three different model set-ups: a simulation where chemical reactions are not considered (passive tracer), one where only the $NO_x - O_3$ cycle is simulated (Photostationary steady state), and the last one where the reactions involving VOC are also implemented (a Complex Chemical Mechanism reduced for the CFD model). With normalized concentration the source emission dependency is dismissed, and the deviation with respect to tracer is only due to the chemical reactions of pollutants with each other. Overall, in either case, the chemical reactions change the NO and $NO_2$ concentration in relation to be regarded as non-reactive species.

As it can be seen in this study, the solar position and the available $O_3$ in the canopy determine the impact of the chemical reactions on NO and $NO_2$ concentrations. Additionally, the influence of wind speed is evaluated in order to establish a relation of [NO] and $[NO_2]$ and the $[O_3]$ within the canopy. The wind speed defines the vertical transport in the domain and consequently, the inflow of $O_3$ into the street. Based on this, the higher differences on $[NO]_N$ and $[NO_2]_N$ from the tracer are obtained in the higher $[O_3]$ of background case with faster wind speed. That is due to a major vertical exchange of concentration that intensifies the $NO_x$ removal and the $O_3$ entrainment in the street. However in the representative case of winter when the amount of $O_3$ is generally lower, the variation of wind speed has hardly influence on $[NO]_N$ and $[NO_2]_N$. Focusing on analyzing the conditions as from which the differences between chemical scenarios are notable, it is essential to incorporate the VOC emission because of traffic exhausts. To quantify how affect the VOC levels on [NO] and $[NO_2]$ within the street, two realistic emission ratios of $VOC/NO_x$ are considered ($VOC/NO_x$=1/2 and $VOC/NO_x$=1/5). The vertical fluxes of concentration and the average con-





centration below the canopy under several conditions show the deviation on $[NO]_N$ and $[NO_2]_N$ by including VOC reactions against to the $NO_x - O_3$ system. The joint evaluation of all parameters conclude that the largest variation among chemical scenarios are found with high available ozone in the street and it is intensified by increasing the amount of VOC within the street. The lower wind speed conditions implies that the VOC concentration holds retained within the canopy and by means

of chemical reactions increases the NO conversion into $NO_2$. Likewise, the formation of $O_3$ related to the photolysis of $NO_2$ along with the $O_3$ concentration of background rise the difference on the $[NO_2]_N$ considering the VOC reactions with respect to the simple chemical mechanism.

Nowadays, the big concern about air quality in big cities is carrying out many studies about dispersion of the primary pollutants related to traffic emissions. That focuses attention on $NO_2$ since it is the most detrimental for citizens health. To

simulate its dispersion in a real urban area is a big challenge from the perspective of modeling at microscale with a CFD model. To obtain the best compromise between accuracy and the CPU time required in order to simulate the $NO_2$ dispersion within the streets, this work seeks the chemical reactions needed as a function of several atmospheric conditions. Therefore, in representative conditions of winter when the $O_3$ concentration is usually small and the solar radiation is low, the implementation of $NO_x - O_3$ reactions can be suitable with slight errors. Even with a low wind speed, the behavior of $[NO]_N$ and $[NO_2]_N$ in

average is quite close to that tracer. In contrast when the amount of the available $O_3$ within the street increases, the differences linked to the chemical reactions considered rises. The non linearity of chemistry induces that $[NO]$ and $[NO_2]$ are not inversely proportional with respect to wind speed, unlike the non-reactive pollutant. Moreover the VOC levels within the street is also an important factor to take into account. With conditions of high background $O_3$ concentration and with an emission ratio of $VOC/NO_x = 1/2$, the VOC chemical reactions seem to be necessary in order to reproduce the NO and $NO_2$ dispersion in the

streets. These assumptions can be useful and provide information to study the pollutants dispersion in real urban areas using a CFD model.

*Author contributions.* B. Sanchez performed the CFD simulations. B.Sanchez, JL. Santiago and A. Martilli discussed the results from the CFD simulations and the conclusions. M. Palacios reviewed the chemical comments. F. Kirchner developed the complex chemical mechanism implemented in the CFD model. B. Sanchez prepared the manuscript with contributions from all co-authors.

*Acknowledgements.* This study has been supported by European Project LIFE MINOx-STREET (LIFE12 ENV/ES/000280) funded by EU. Authors thank Extremadura Research Centre for Advanced Technologies (CETA-CIEMAT) by helping in using its computing facilities for the simulations. CETA-CIEMAT belongs to CIEMAT and the Government of Spain and is funded by the European Regional Development Fund (ERDF).



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



**Table 1.** The complex chemical mechanism (CCM-CFD): The units of the rate constants of first order reactions are $s^{-1}$, of second order reactions $cm^3 s^{-1}$. The rate constants for the reactions called "thermal" can be calculated by $k = A * \exp[(-E/R)/T]$, of reactions called "Troe" by $k = \{k_0(T)[M]/(1+(k_0(T)[M]/k_\infty(T)))\}0.6^{\{1+[\log_{10}(k_0(T)[M]/k_\infty(T))]^2\}^{-1}}$, and reactions called "Troe-equilibrium" by $k = A\exp(-B/T) \times \{k_0(T)[M]/(1+(k_0(T)[M]/k_\infty(T)))\} \, 0.6^{\{1+\log_{10}[k_0(T)[M]/k\infty(T)]^2\}^{-1}}$, where $k_0(T) = k_0^{300}(T/300)^{-n}$ and $k_\infty(T) = k_\infty^{300}(T/300)^{-m}$.

| | | |
|---|---|---|
| $NO_2$ | $NO + O_3$ | photodissociation |
| $O_3$ | $O^1D$ | photodissociation |
| HCHO | CO | photodissociation |
| HCHO | $CO + 2\,HO_2$ | photodissociation |
| ALD | $CO + HO_2 + MO2$ | photodissociation |
| $O^1D + N_2$ | $O_3$ | thermal(A=1.80E-11, E/R=-110.0) |
| $O^1D + O_2$ | $O_3$ | thermal(A=3.20E-11, E/R=-70.0) |
| $O^1D + H_2O$ | 2 HO | thermal(A=2.20E-10, E/R=0.0) |
| $HO + NO_2$ | $HNO_3$ | troe($k_0^{300}$=2.60E-30, n=3.2,$k_\infty^{300}$=2.40E-11, m=1.3) |
| $HO_2 + NO$ | $NO_2 + HO$ | thermal(A=3.70E-12, E/R=-250.0) |
| $O_3 + NO$ | $NO_2$ | thermal(A=2.00E-12, E/R=1400.0) |
| $HO + SO_2$ | $HO_2$ | troe($k_0^{300}$=3E-31, n=3.3, $k_\infty^{300}$=1.50E-12, m=0.0) |
| CO + HO | $HO_2$ | K = 1.5d-13*(1.+2.439e-20*airc) |
| ALK + HO | 0.931 XO2 + 0.842 $HO_2$ + 0.011 CO + 0.011 HO + 0.019 HCHO + 0.051 MO2 + 0.378 ALD + 0.096 XO2N | thermal(A=8.05E-12, E/R=237.0) |
| OLE + HO | 1.001 XO2 + 0.998 $HO_2$ + 1.011 HCHO + 0.002 ACO3 + 0.747 ALD | thermal(A=5.69E-12, E/R=-474.5) |
| ARO + HO | 0.950 XO2 + 0.950 $HO_2$ + 1.860 ALD + 0.050 XO2N | thermal(A=5.35E-12, E/R=-355.0) |
| ALD + HO | ACO3 | thermal(A=5.55E-12, E/R=-331.0) |
| HCHO + HO | $HO_2 + CO$ | thermal(A=1.00E-11, E/R=0.0) |
| $OLE + O_3$ | 0.344 $HO_2$ + 0.383 CO + 0.303 HO + 0.135 XO2 + 0.682 HCHO + 0.092 MO2 + 0.007 ACO3 + 0.630 ALD | thermal(A=1.28E-15, E/R=907.1) |
| $NO_2 + ACO3$ | PAN | troe($k_0^{300}$=9.70E-29, n=5.6,$k_\infty^{300}$=9.30E-12, m=1.5) |
| PAN | $NO_2 + ACO3$ | troe-equil ($k_0^{300}$=9.70E-29, n=5.6, $k_\infty^{300}$=9.30E-12, m=1.5, A=1.16E28, B=13954.) |
| XO2 + NO | $NO_2$ | thermal(A=4.00E-12, E/R=0.0) |
| MO2 + NO | $NO_2 + HO_2 + HCHO$ | thermal(A=4.20E-12, E/R=-180.0) |
| ACO3 + NO | $NO_2$ + 0.046 $HO_2$ + 0.046 CO + 0.954 MO2 | thermal(A=2.00E-11, E/R=0.0) |
| XO2N + NO | 0.939 ONIT | thermal(A=4.45E-12, E/R=-39.9) |





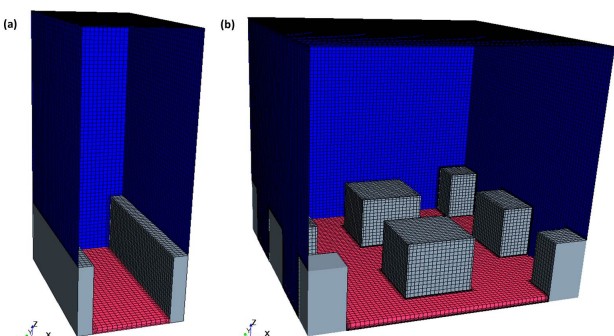

**Figure 1.** Illustration of the computational domains: (a) a single street canyon with $H$=16 m and $W$=16 m (2D-geometry), and (b) staggered array of cubes with $H$, $W$ and $L$ equal to 16 m (3D-geometry).

**Table 2.** Summary of the simulations performed in 2D and 3D geometries. All chemical scenarios are simulated considering the combination of the two wind speed and two background ozone concentration cases.

| Chemical Scenarios | Wind Speed | Background $[O_3]$ |
|---|---|---|
| Non-reactive | $u_\tau$ | High $[O_3]$ |
| Photostationary Steady State | $u_\tau$ | Low $[O_3]$ |
| CCM-CFD (VOC/NO$_x$=1/5) | $u_\tau/2$ | High $[O_3]$ |
| CCM-CFD (VOC/NO$_x$=1/2) | $u_\tau/2$ | Low $[O_3]$ |

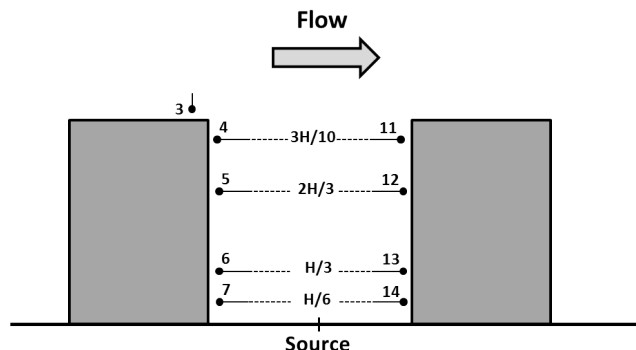

**Figure 2.** Location of the comparison points with experimental measurements of Meroney et al. (1996)





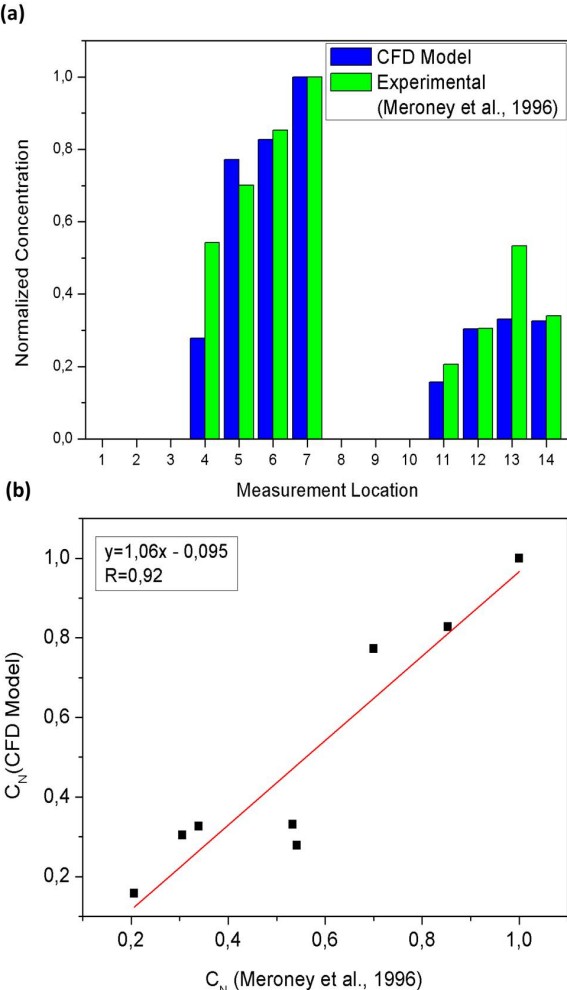

**Figure 3.** Comparison between simulation results and experimental data (Meroney et al., 1996). (a) Comparison between modeled concentration and experimental measurements at each location. (b) Linear fit between experimental measurements and modeled concentrations.

**Table 3.** Statistic metrics obtained for the validation of CFD results with experimental data (Meroney et al., 1996) and the acceptance criteria proposed by (Chang and Hanna, 2005) for urban configuration.

|  | Acceptance Criteria | Statics Values |
| --- | --- | --- |
| NMSE | < 1.5 | 0.03 |
| FB | (-0.3,0.3) | 0.068 |
| R | >0.8 | 0.92 |



**Table 4.** Values of Normalized mean square error (NMSE), fractional bias (FB), correlation (R) and the maximum concentration error with respect to concentration results from box model simulation.

|        | NMSE              | FB                 | R    | ERROR (%) |
|--------|-------------------|--------------------|------|-----------|
| NO     | $4{,}33 \ 10^{-06}$ | $-1.97 \ 10^{-03}$ | 0,99 | 0.28      |
| $NO_2$ | $1{,}53 \ 10^{-06}$ | $1.20 \ 10^{-03}$  | 0,99 | 0.18      |
| $O_3$  | $1{,}94 \ 10^{-07}$ | $3.77 \ 10^{-04}$  | 0,99 | 0.085     |

**Table 5.** Quadratic relative differences of concentration different time step ($\%$).

|        | $QD(t_s = 0.1s)$ | $QD(t_s = 2s)$ |
|--------|------------------|----------------|
| NO     | 0.0061           | 0.023          |
| $NO_2$ | 0.0062           | 0.022          |
| $O_3$  | 0.0014           | 0.0055         |

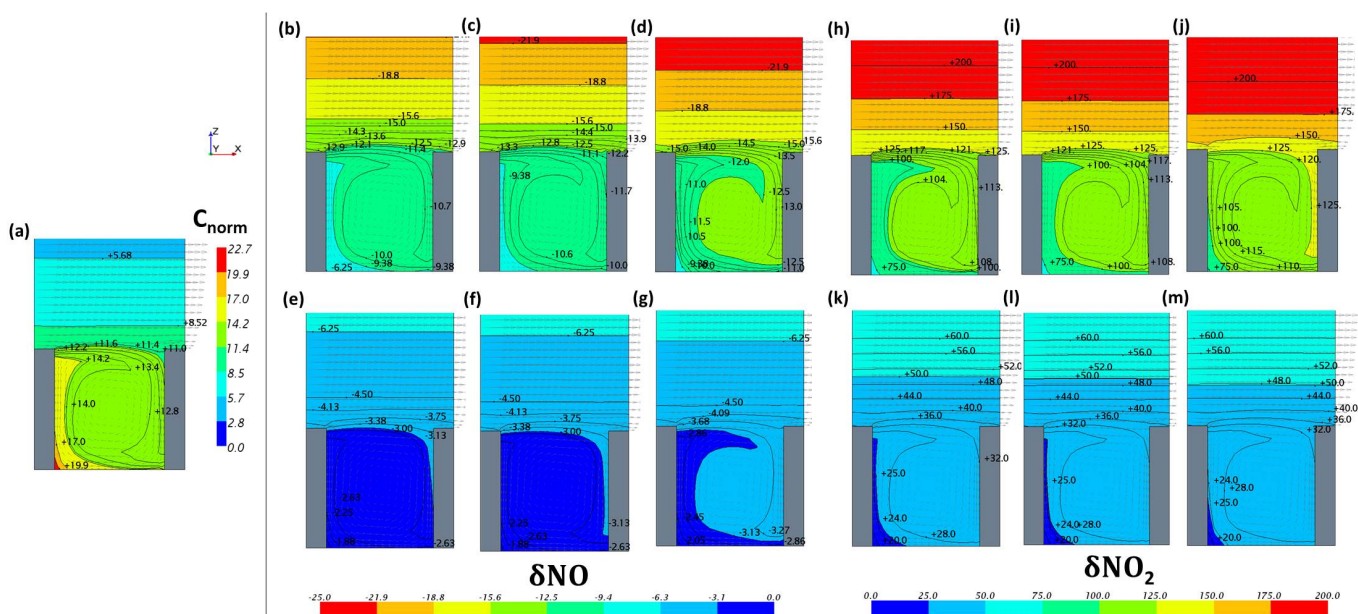

**Figure 4.** In the 2D-geometry: (a) Distribution of the normalized concentration of tracer and the $\delta NO$ ($\delta NO_2$) for the PSS, CCM1/5 and CCM1/2 in the lower velocity case are depicted with higher $O_3$ in b-d (h-j) and with lower $O_3$ in e-g (k-m) respectively.





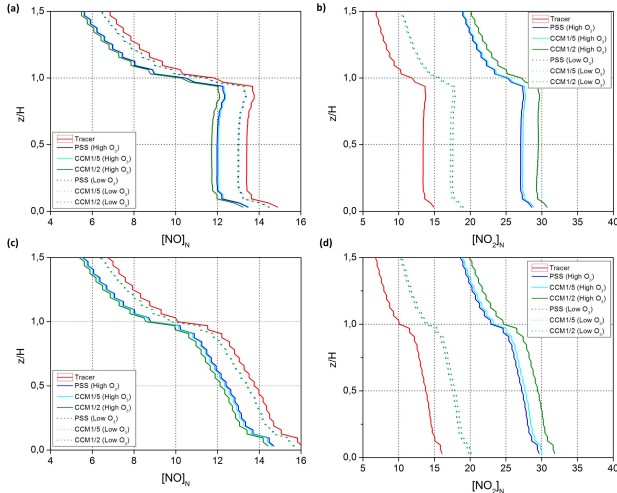

**Figure 5.** Vertical profiles of horizontal spatial average of $[NO]_N$ and $[NO_2]_N$ obtained with lower velocity for the different chemical mechanisms with high $O_3$ (dotted line) and low $O_3$ (solid line) in 2D (a,b) and 3D (c,d) geometries.

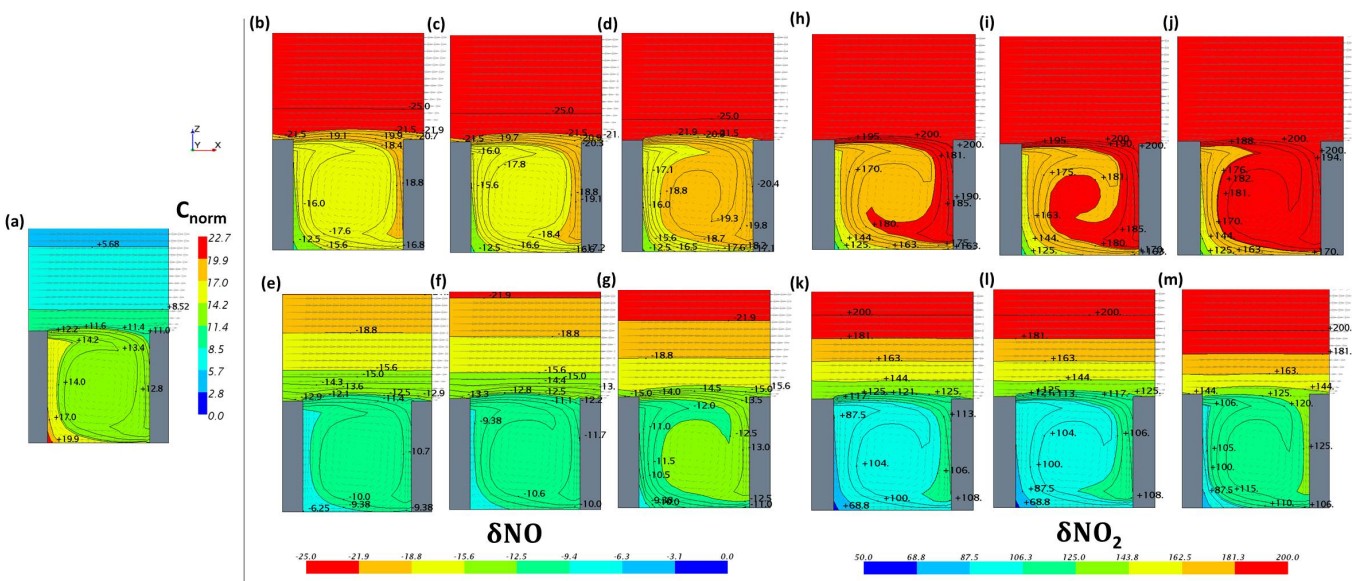

**Figure 6.** In the 2D-geometry: (a) Distribution of normalized concentration of tracer and the $\delta NO$ ($\delta NO_2$) for the PSS, CCM1/5 and CCM1/2 in the high $O_3$ case are depicted with the faster wind speed in b-d (h-j) and with slower wind speed in e-g (k-m) respectively.





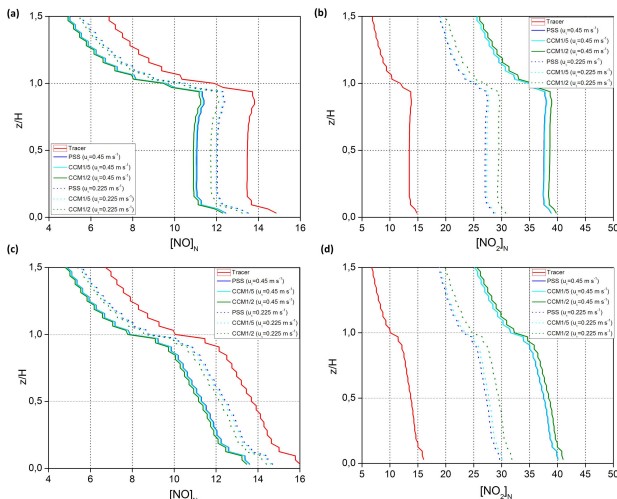

**Figure 7.** Vertical profiles of horizontal spatial average of $[NO]_N$ and $[NO_2]_N$ obtained with high $O_3$ for the different chemical mechanisms with the slower (dotted line) and faster (solid line) wind speed in 2D (a,b) and 3D (c,d) geometries.

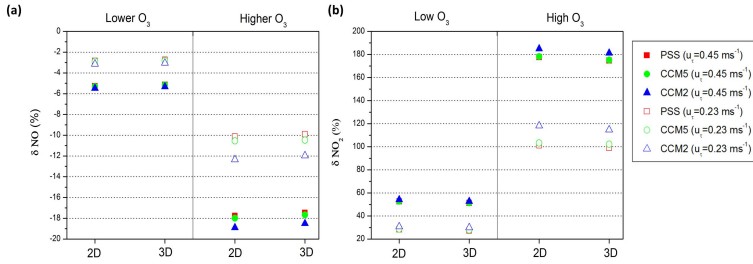

**Figure 8.** Deviation $[NO]_N$ and $[NO_2]_N$ in average below the canopy with respect to its corresponding normalized concentration averaged of the tracer in 2D and 3D geometry.



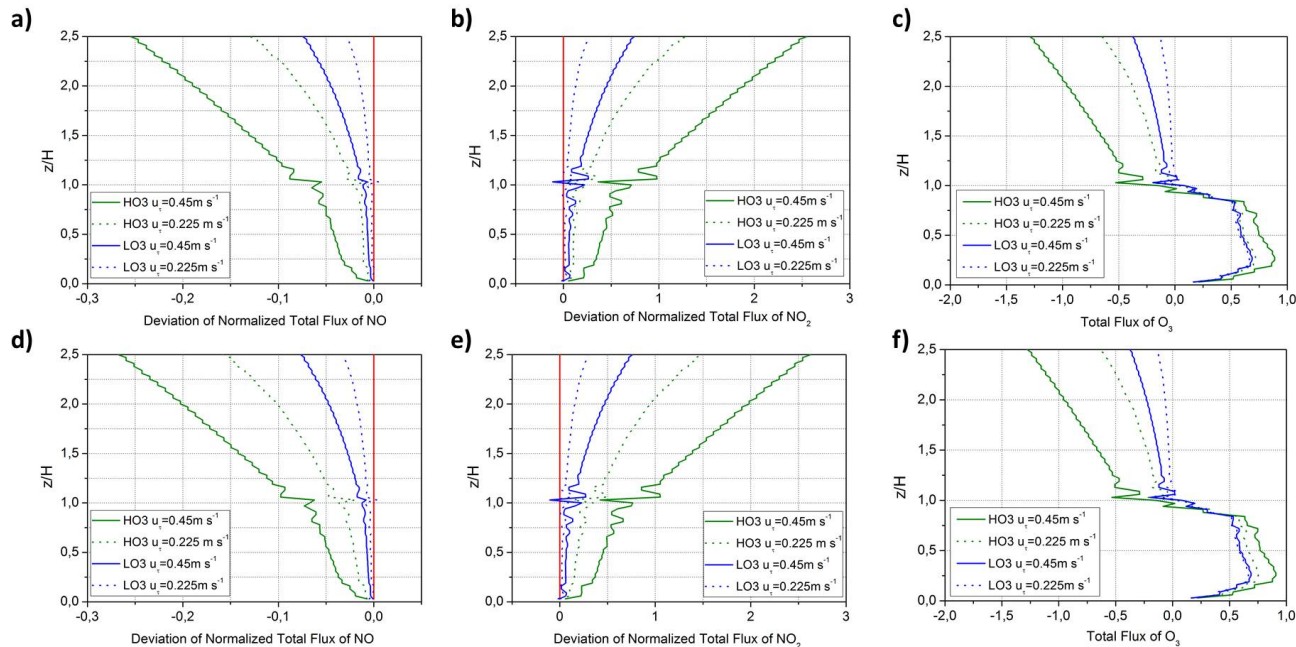

**Figure 9.** Vertical profiles of the the deviation of $< \overline{w\,NO} >_N$ and $< \overline{w\,NO_2} >_N$ from the normalized total flux of tracer and the $< \overline{w\,O_3} >_N$ for the (a-c) PSS and (d-f) CCM1/2 in the 3D geometry for all study cases. The red line is the normalized total flux of the tracer, and the green and blue lines represent respectively the high $O_3$ and low $O_3$ cases, whereas dotted and solid lines the faster and slower wind speed cases respectively.





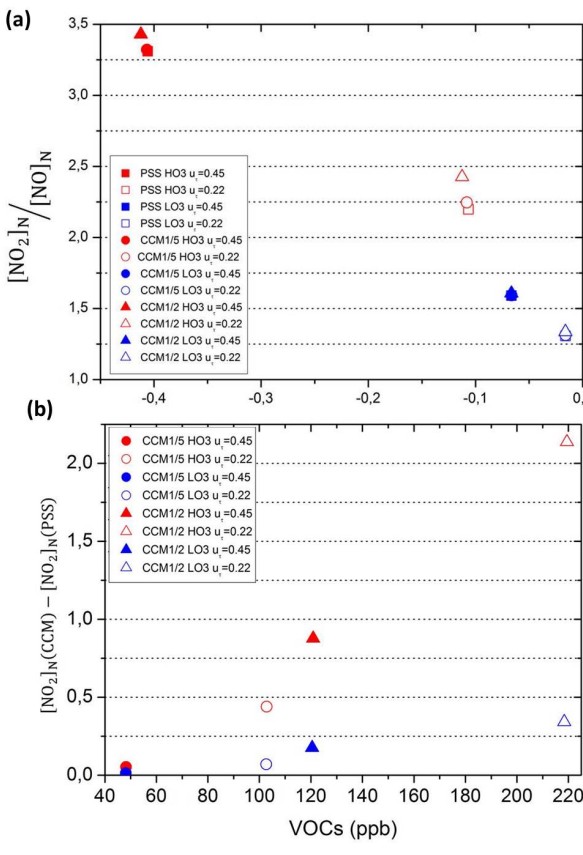

**Figure 10.** (a) The average concentration within the canopy of $[NO_2]_N/[NO]_N$ against the total flux of $O_3$ at roof level and (b) the variation of the average concentration within the canopy of $[NO_2]_N$ with CCM1/5 and CCM1/2 from that the PSS over the average concentration of VOC.