# Peer review of "CFD Modeling of Reactive Pollutants Dispersion in Simplified Urban Configurations with Different Chemical Mechanisms"

_Atmospheric Chemistry and Physics, 2016_

## Referee Comment (RC1) · Anonymous Referee #2 · 29 Jun 2016

GENERAL COMMENTS The paper entitled "CFD Modeling of Reactive Pollutants Dispersion in Simplified Urban Configurations with Different Chemical Mechanisms" deals with the modeling of reactive pollutants using a computational fluid dynamics (CFD) model. Three chemical approaches are considered. Main results show that the presence of ozone in the street acquires an important role in NO and NO2 dispersion.

The paper is an application of a CFD model to the study of dispersion of reactive pollutants in idealized geometries. Chemical schemes are implemented in the CFD model and this is an important and novel contribution since most studies focused on non-reactive pollutants. Overall the authors have done a lot of technical and scientific work, but in my opinion it is presented in a slightly confused way. It is not obviously

clear which cases have been studied and main results obtained. The description of the cases is not done in a schematic way and this makes the paper, at least in the methodology sections, hard to understand. Also some sentences are not scientifically sound. More attention should be paid to what is discussed and introduced based on previous literature studies and references should be given. Also English should be checked. I suggested several corrections, but I cannot guarantee everything has been checked so I suggest the paper to be checked by a native speaker. Further, there is no attempt to analyse physical processes behind the results. The above and below issues should be addressed by the authors before publication. The authors have the necessary expertise to address all the issues which concern mainly the presentation and analysis of results and not the methodology employed which is of high scientific level.

SPECIFIC COMMENTS Abstract. - Overall I found the abstract a little bit confused Several information are mixed without a proper organization. I suggest the authors to introduce the problem, then discuss the methodology in a schematic way and then main results. - The sentence "it is reduced to 23 species . . .." Is not clear since the reader at this stage of the paper may not know the starting point from which the reduction takes place. - "the concentration of reactive pollutants is affected by many atmospheric parameters" is too generic. Which parameters? And which are you considering here? - "role in NO and NO2 concentration" and not dispersion - "The joint evaluation of both parameters": what is "both" referred to?

Introduction. - Please check throughout the paper that NOx, NO2 etc. have been defined before using chemical formula. - define UCL - Pag. 2 Line 28. This is a repetition of what already mentioned at line 24 - Pag. 3 Lines17-18. This concept is a repetition of what already mentioned at line 5 of pag. 2. Overall also the introduction is slightly confused. The same concepts are repeated throughout the section. - Pag. 3 Line 19. How did you estimate "a factor of 2"? - At the end please underline the structure and the original contribution of the paper

Chemical Schemes used. - RACM? - Reduced to 25 reactions with respect to...? Please introduce the scheme otherwise it is not easy to understand - Overall also this section is hard to follow. As in the abstract the cases investigated are mixed, while they should be presented in a more schematic way avoiding to first introduce, then discuss of a scheme, them move to the other and coming back to the first and so on... - It is not clear why two so close wind velocity have been chosen. They are both calm conditions. If the results achieved are different based on the velocity, this should be discussed in terms of flow regime in the streets and turbulence. - How did you estimate 930 vehicles?

CFD model evaluation. Please remove the sentence "despite the fact that ... was previously validated..." This is not scientifically sound. The present paper is based on CFD simulations and thus the simulations should be critically validated. For this reason, I suggest also to validate flow and turbulence obtained from CFD and not only concentration at few points. This would add value to the paper and justify the accuracy of results. - Did the authors performed any sensitivity test of the grid and domain size? - Remarks should be added about other turbulence modelling, such as the RSM, LES

Results. - Please discuss also the physical processes for 2d and 3d geometries and discuss throughout the text and in the conclusions also the differences between 2d and 3d geometries. It is not clear if one introduce more errors simplifying the geometry or the chemical reactions. - Pag. 9 Lines 14-15. What do you mean with "apparently"? It is recognised that 3d geometries lead to corner vortices which should improve the dispersion from the streets. Please adjust and add a reference. Also he sentence "Therefore, the residence time of each reactive compound within street is determined by building configurations and wind speed " is not clear since this is true both for 2d and 3d. - Pag. 9 Line 18. Domains? Do you mean in 2d and 3d geometries? - Pag. 11 Line 6. It is not strictly true that the concentration of non-reactive pollutant is inversely proportional to wind speed. It is true in flat terrain, but in the streets it may depend on other variables. Please add a reference. - Pag. 11 Line 28. "The effect of

considering reactive pollutants is enlarged.." You mean the importance of considering reactive pollutants for a better model accuracy...? Please check this kind of sentences throughout the paper.

---

## Referee Comment (RC2) · Anonymous Referee #1 · 14 Jul 2016

In the paper, Sanchez et al. study how NOx-Ozone-VOC chemistry over a city can be best simulated, accounting for reliability and computer time. The paper describes three general approaches: a setup where NOx and Ozone are passive tracers (considering that transport and mixing in street canyons are relatively compared with chemistry), a setup with simple chemistry (the photostationary state, NO2 + O2 + hv <-> NO + O3, computationally efficient, most relevant chemical processes), and a setup with an expanded set of chemical equations derived from the RACM scheme. The authors test the setup in an idealized 2D and 3D representation of a (number of) street canyons, with high and low zenith angle and corresponding solar radiation intensity (winter and summer situations, or low and high background Ozone concentrations), and VOC/NOx

emissions rates of 1/5 and 1/2. Furthermore, they test the influence of wind speed (u = 0.45 m.s-1 and u=0.23 m.s-1). The authors thus choose to manipulate a large number of input variables, which results in a large number of different results. This is both a strong and weak point of the paper. It is strong because the results are put in a broad perspective. It is weak because it is difficult to keep track of all the different results. I suggest that the authors make an effort to create a better overview of the different experiments, and possibly introduce acronyms for them, and use them as titles in the figures and tables. Nevertheless, the authors come to clear and understandable conclusions:

- Wind speed determines the vertical exchange rate, and therefore the influx of background Ozone.

- The choice of chemical scheme is more important in the situations with high O3 concentrations

- In high wind and high background O3 concentrations, it is essential to use an expanded set of chemical equations. The impact obviously increases with the VOC emission rate.

In my opinion, the paper describes a very well-organised set of experiments, which yield clear conclusions. Although the experiments already encompass a large number of manipulations, the conclusions remain somewhat qualitative (in conditions of high/low Ozone, wind speed, VOC emission rates). As a result, the results will be difficult to apply directly by other researchers.

It would be convenient if the authors could work towards specifying threshold values for those variables or describing the sensitivity of the results to these variables. I understand that this would involve doing the same experiments with a broader range of the variables and/or more and smaller step sizes. Probably this also includes using non-idealized street geometry. I am not suggesting that the authors do all that for the current paper. But they should include clear recommendations on how to turn the

qualitative conclusions into quantitative and applicable results.

I recommend publication of this paper in Atmospheric Chemistry and Physics with minor revisions: 1) create a better overview over the many different experiments and 2) include recommendations on how to come to quantitative and applicable results. The paper also needs English grammar editing.

Detailed comments

- 'researches': research is never a plural in English. Better use 'studies'.

Abstract

- line 12: explain the role of wind speed

- line 14: founded > found

- line 14: related with > related to

- line 16-17: rephrase

- line 19: '. . . we found IT more . . .'

- line 29: '.. in urban areas IT becomes more . . .'

- line 33: 'marked'? Not clear what you mean

- line 35: 'O3 sensitivity', do you mean the sensitivity of O3 to ??? or the sensitivity of the results to O3? This is not clear.

Page 3:

- line 3: '. . . whereby increasing the wind speed enhances the exchange with the overlying air. . .'

- line 3: strengthens

- line 23: 'The aim of this work is to determine in which conditions it is essential . . .'

Page 4, line 8: JNO2 and k are not shown in the equations

Page 5:

- Line 11: 'As well. . .' > 'Also. . .'

- Line 25-end: rephrase

Page 6:

- Section 3.2: restructure this section. It is now not very well organised. It sounds like new experiments are introduced continuously with another set of manipulated variables. I think it would be good to introduce a table with an overview over all the different experiments right in the beginning, and then use this section to explain the individual experiments and their mutual relationships. I found it somewhat confusing that you refer to the low/high background O3 concentration experiments as winter/summer and high/low zenith angle experiments (I assume these are the same experiments). You might also consider introducing descriptive acronyms for the experiments, which you could use in the titles of the figures and tables and in the text.

- Line 17: are the symmetric conditions also applied to the concentrations?

- Line 21: 'within the typical range of values. . .'

- Line 23: 'two background O3 concentrations': specify which, this has not yet been introduced at this stage. Page 7:

- Line 4-5: the NO/NO2 concentrations also depend on the intensity of turbulent mixing in the boundary layer, e.g. the difference between clear and cloudy days, or strong/weak inversion. How has this been addressed?

- Line 21: 'from a well-established chemical box model': describe which one in some words. After reviewing the paper I do not have a clear idea on where the results of this

box model are used. Is this an essential part of the paper?

Page 9:

- Line 13: I found the use of the word 'canopy' confusing. E.g. in page 10, line 11 you write 'below the canopy'. This suggests that you simulate concentrations below trees. Is this true? Please clarify. How tall is the canopy?

- Line 14: '. . ., and the pollutants reside longer in the street. . .'

- Line 18: 'To facilitate the comparison. . .'

- Line 19: is the source emission rate Q expressed per unit area? Otherwise the normalisation with A_em is not needed.

- Line 20: It would make sense to use C_N instead of Cnorm, because later on you use [NO2]N etc.

- Line 20: describe L

Page 10:

- Line 4: 'the effect of . . . emission ratio settings ON the amount. . .'

- Line 5: '. . . fixed emission rates. . .'

- Line 11-13: 'On the other hand. . .' suggest that the new sentence is in contradiction with the previous one, but this is not the case.

- Could you explain why in the low O3 case the VOC emissions do not make much difference?

Page 11, line 11 to end: this is discussion, not results. I would expect that you describe the difference between the 2D and 3D experiments here.

Page 12:

- Line 4: on accurate way > accurately.

- Line 9: 'without hardly differences' is a double negative: 'with hardly any differences' or 'without any differences' or 'without large differences'

Page 13:

- Line 14: slightly minor/major: I think you mean slightly smaller or larger?

- Line 25: IN this way

- Line 33: reacts > react

Page 14: Line 2-4: rephrase

Page 15:

- Line 4-5: rephrase

- Line 6: rise > increase

- Line 8-20: Can you make this part more quantitative, and include recommendations to direct further research?

- The subject of computation time is not addressed here. Perhaps implicitly, but not explicitly.

Table 2: specify the background concentrations (instead of high/low)

Figures: The fonts in the figure titles/axes/legends are extremely small.

Figure 6: subplot e should be the same as 4e? They refer to the same experiment if I understand it correctly.

---

## Author Comment (AC1) · 2 Sep 2016

Thank you for your assessment about the manuscript and your comments. The manuscript has been modified to create a better overview of all cases considered in this work and so, providing some quantitative conclusions from the results. All typing remarks have been modified in the revised manuscript.

The responses to the SPECIFIC COMMENTS are described in the following lines and the corresponding changes in the manuscript have been highlighted in blue in the PDF file:

[Figure]

***Abstract***. *Line 16-17: rephrase*

Please see this section in the PDF file.

***Introduction.*** *Line 33: 'marked'? Not clear what you mean. Line 35: 'O3 sensitivity', do you mean the sensitivity of O3 to ??? or the sensitivity of the results to O3? This is not clear.*

It is referred as the sensitivity of O3 to the NOx emission level, since they evaluated the changes in O3 concentration on varying the NOx and VOC emission levels. It has been clarified in the revised manuscript, please see PDF file (Page 2, lines 24-27).

***Model Description (Section 3.1)***. *Page 5, line 25-end: rephrase*

Please see the PDF file (Page 5, lines 18-22).

***Simulation Setup (Section 3.2)*** *restructure this section.*

This section has been re-organized as per your remarks. Please see this section in the PDF file.

*Page 6, line 17: Are the symmetric conditions also applied to the concentrations?*

For pollutants concentration, the symmetric conditions are only imposed in y-direction. The outlet condition was established at the top of the domain and a constant value of concentration was imposed for each pollutant. Please see the PDF file (Page 6, lines 23-25).

*Line 4-5: The NO/NO2 concentrations also depend on the intensity of turbulent mixing in the boundary layer, e.g. the difference between clear and cloudy days, or strong/weak inversion. How has this been addressed?*

We agree on this with the reviewer. However, the influence of the turbulent mixing intensity at the boundary layer on NO/NO2 concentration is not addressed in this paper. The same type of turbulent mixing (induced only by the interaction between the wind and the presence of buildings without any thermal effect) at a peak hour of traffic emissions is considered in all scenarios.

**CFD Model Evaluation:** *(Page 7, line 21):'from a well-established chemical box model': describe which one in some words. After reviewing the paper I do not have a clear idea on where the results of this box model are used. Is this an essential part of the paper?*

This section is important in this paper since it evaluates the implementation of the chemical terms in the CFD model. To achieve this, the CFD is run as box model (e. g. without transport and diffusion) and the results of pollutants concentrations were compared with the outcomes derived with the chemical box model used to test the chemical mechanism.

This issue has been clarified. Please see the PDF file (Page 8, lines 8-18).

**Results.** *Page 9, line 13: I found the use of the word 'canopy' confusing. E.g. in page 10, line 11 you write 'below the canopy'. This suggests that you simulate concentrations below trees. Is this true? Please clarify. How tall is the canopy?*

What is meant in the text, in reality is below canopy top, or within the canopy. In this work, the height of the canopy is regarded as that of buildings (H). This concept has been clarified in the revised manuscript (Please see the PDF file, page 9, lines 3-4).
*Page 9, line 19: Is the source emission rate Q expressed per unit area?*

Q is the emission source rate and it is expressed in kg s-1, so that Q/Aem is the emission density flux, in Kg/s/m2. We have noticed that there was a mistake in the expression for the normalized concentration. The normalized concentration is now:

$$C_N = \frac{C \, u_\tau \, A_{Em}}{Q} \tag{1}$$

Please see the PDF file (Equation 8; Page 9, line 12).

*Could you explain why in the low O3 case the VOC emissions do not make much difference?*

In the low O3 case, the selected zenith angle is representative of winter conditions, therefore the photolysis constants are smaller than that given in a representative summer case. This implies that the background O3 concentration computed by the photochemical equilibrium equation is low. In turn, the VOC oxidation cycle is also limited by the photolysis of some VOCs. Besides, given that the NOx emissions dominate over the VOC emissions, the high levels of NOx concentration tend to inhibit O3 formation and limit the O3 production through VOC reactions. Therefore, the difference in NO and NO2 concentration after including VOC reactions against the photostationary steady state is negligible in this low O3 case. So the weak dependence on VOC concentration is a combination of the low O3 and small photolysis rates.

We have included this text in the revised manuscript. Please see the PDF file (Page 10, lines 16-23)

*Page 11, line 11 to end: this is discussion, not results. I would expect that you describe the difference between the 2D and 3D experiments here.*

This section has been modified in order to show more results. Please see the PDF file (Page 10-11).

In this work, the differences between 2D and 3D geometries were not addressed since it would be a complex study in order to carry it out here as well. In this section, the influence of several wind speed on NO and NO2 dispersion in the streets is studied in different types of geometries. The objective is to understand whether the deviation on normalized concentrations from that of the tracer by including chemical reactions had the same behavior or not in both geometries. Note that the concentration of a non-reactive pollutant is inversely proportional to wind speed (Parra et al., 2010), but the chemical reactions modified this behaviour. Finally, the same conclusions, in terms of chemical effects, were obtained from the CFD results for both geometries.

*Page 14: Line 2-4: rephrase.*

Please see the PDF file (Page 13, lines 22-23).

*Page 15. Line 4-5: rephrase.*

Please see the PDF file (Page 14, lines 22-23).

*Page 15. Line 8-20: Can you make this part more quantitative, and include recommendations to direct further research?*

This part has been modified in order to provide more quantitative information from the experiments performed in this work. Please see the PDF file (Page 14, line 30 to end)

*The subject of computation time is not addressed here. Perhaps implicitly, but not explicitly.*

We agree with the referee's comment, since we have not evaluated explicitly the differences in computational load on using one chemical scheme or other one. But it is implicitly included in our conclusions, because the aim of this work is to optimize the modeling of NO and NO2 dispersion and so as to reduce the computational time required to carried it out.

We have indicated the relationship between computational time needed to simulate the complex chemical scheme and the photostationary steady state (Page 3, lines 17-19)

*Figure 6: subplot e should be the same as 4e? They refer to the same experiment if I understand it correctly.*

The figures for the same simulations are Figure 6 (e-g) and Figure 4 (b-d) corresponding to the case of O3=39.9 ppb with $u_\tau$=0.23 m s$^{-1}$ (Case 2).

Please also note the supplement to this comment:
http://www.atmos-chem-phys-discuss.net/acp-2016-202/acp-2016-202-AC1-supplement.pdf

[Figure]

**Supplement:**

[revised manuscript text omitted]

---

## Author Comment (AC2) · 2 Sep 2016

Thank you for your appraisal of the manuscript and your comments for improving it. In regards to the information which was rather confusing, we have enhanced the comprehension. Particularly, we have clarified the methodology used and highlighted the main results from this work.

The responses to the SPECIFIC COMMENTS are described in the following lines and the corresponding changes in the manuscript have been highlighted in blue in the PDF file:

*Abstract*. *Overall I found the abstract a little bit confused. Several information are*

[Figure]

*mixed without a proper organization. I suggest the authors to introduce the problem, then discuss the methodology in a schematic way and then main results.*

This section has been re-organized and modified the confused sentences according to your comments. Please see this section in the PDF file.

*Introduction. Please check throughout the paper that NOx, NO2 etc. have been defined before using chemical formula. Overall also the introduction is slightly confused. The same concepts are repeated throughout the section. At the end please underline the structure and the original contribution of the paper.*

This section has been re-written according to your comments. Please see the PDF file.

*How did you estimate "a factor of 2"?*

This is referred to as the computational time required to carry out the simulation with the complex chemical scheme implies more than twice the time demanded with the use of the photostationary steady state. This has been clarified in the revised manuscript, please see the PDF file (Page 3, lines 17-19).

*Chemical Scheme used. Please introduce the scheme otherwise it is not easy to understand. RACM? - Reduced to 25 reactions with respect to?*

The reduced complex chemical scheme used in this work consists of 23 chemical species and 25 chemical reactions. This chemical scheme was developed from the Regional Atmospheric Chemistry Mechanism (RACM) with 77 species and 237 reactions. Thus, the computational time of simulating the reactive pollutant with the reduced mechanism was also reduced.

The chemical scheme has been introduced in order to understand it easily. Please see the PDF file (Page 3, lines 27-29; Page 4, lines 4-12).

*Simulation Setup. Overall also this section is hard to follow.*

This section has been modified and re-structured as per your recommendations. Please read this section in the PDF file.

*It is not clear why two so close wind velocities have been chosen. They are both calm conditions. If the results achieved are different based on the velocity, this should be discussed in terms of flow regime in the streets and turbulence.*

The reference velocity ($u_\tau$) used in the manuscript represent the friction velocity in order to compute the pressure gradient imposed on cyclic boundary conditions. For $u_\tau$=0.22 m s$^{-1}$ the corresponding wind speeds at 1.5H are 1.9 m s$^{-1}$ and 1.5 m s$^{-1}$ in the 2D and 3D geometries respectively, and the double for the $u_\tau$=0.45 case, so differences are significant in terms of wind speed at 1.5H.

Wind speed and turbulence are proportional to $u_\tau$ and $u_\tau^2$ respectively because thermal effects were not taken into account. Therefore, the flow regime in the streets is the same in both cases.

Some comments have been included in the manuscript in order to clarify this issue. Please see the PDF file (Page 6, lines 8-9).

*How did you estimate 930 vehicles?*

930 vehicles per hour can be representative of medium traffic (Baker et al., 2004). In this way, we have used the same emission source rates ($S_{NO}$ and $S_{NO_2}$) considered in Baker et al. (2004). Therefore, whether the emission source of NOx is around 130 µg m$^{-1}$s$^{-1}$ ($S_{NO_x} = S_{NO} + S_{NO_2}$) and the NOx emission source per vehicle is around 0.5 g km$^{-1}$ (Baker et al., 2004 and Baik et al.,2007). veh $\;\;\mathrm{s}^{-1} = 130\,\mu g\,m^{-1}\,s^{-1}\frac{veh}{0.5 \cdot 10^3 \mu g\,m^{-1}}$

***CFD Model Evaluation**. I suggest also to validate flow and turbulence obtained from CFD and not only concentration at few points.*

Flow and turbulence corresponding to simulations with the 3D set up and the same CFD model were already validated in other work (Santiago et al., 2008). The flow and turbulence in the array of cubic obstacles has been evaluated using the DNS model results. Vertical profiles of the horizontally averaged wind speed, turbulent kinetic energy and Reynolds shear stress were compared as in Santiago et al. (2008). Similar profiles were obtained for both turbulence models (not shown in the manuscript).

*Did the authors perform any sensitivity test of the grid and domain size?*

In both geometries, we carried out a mesh independence test with a finer grid resolution. In the 2D geometry, we tested two grid resolutions: 0.5 m and 1 m in all directions. The vertical profiles of wind speed and turbulent kinetic energy from each mesh were compared with each other. The obtained results were equivalent and we therefore selected the grid resolution of 1 m. The vertical profile of the spatial average of turbulent kinetic energy is shown in Fig. 1. For the 3D geometry, the grid resolutions were: 0.5 m and 1 m in all directions. The wind speed and turbulence from each mesh resolution were compared and they showed the same results (Fig. 2). This is not shown in the manuscript, please see the attached figures.

As for the domain size, the height of the top was selected based on the results obtained in Coceal et al. (2006). They evaluated the effect of several domain heights concluding that 4H is a good assumption.

We have included some remarks about the sensitivity tests and added the reference in the manuscript. Please see in the PDF file (Page 6, lines 1-5).

*Remarks should be added about other turbulence modelling, such as the RSM, LES.*

Some comments about turbulence modeling have been added (Page 5, lines 11-14):

The CFD model used is based on the Reynolds-averaged Navier-Stokes equations (RANS) with a k-$\varepsilon$ turbulence model. This model allows to evaluate the effect of several parameters using a wide set of simulations within a reasonable CPU time. The turbulence can be solved more accurately by other models such as Large Eddy Simulation or Direct Numerical Simulation, however the CPU load increases considerably and it would limit the number of simulations.

*Results. Discuss also the physical processes for 2d and 3d geometries and discuss throughout the text and in the conclusions also the differences between 2d and 3d geometries. It is not clear if one introduce more errors simplifying the geometry or the chemical reactions.*

The differences between the processes in the 2D and 3D geometries were not addressed here because we just focused on evaluating whether the effects of including chemical reactions led to the same conclusions in different types of geometries. Moreover, we would like to stress that the 2D is not considered a simplification of the 3D, rather a different type of urban structure.

In the course of this work, we have tried to understand how the behavior of a reactive pollutant differs from that of the non-reactive pollutant in each geometry. From the results, we have concluded that under some atmospheric conditions the difference between the chemical scenarios could be important in both geometries, regardless of the type of the geometries used in this study.

This approach has been modified in the revised manuscript in order to clarify the confused information. Please see PDF file.

*It is recognised that 3d geometries lead to corner vortices which should improve the*

*dispersion from the streets. Please adjust and add a reference. Also the sentence 'Therefore, the residence time of each reactive compound within street is determined by building configurations and wind speed' is not clear since this is true both for 2d and 3d. - Pag. 9 Line 18. Domains? Do you mean in 2d and 3d geometries?*

This has been clarified in the PDF file (Page 9, lines 5-9).

*It is not strictly true that the concentration of non-reactive pollutant is inversely proportional to wind speed. It is true in flat terrain, but in the streets it may depend on other variables. Please add a reference.*

Under the assumptions considered in this work, the concentration of non-reactive pollutant is inversely proportional to wind speed as shown in Parra et al. (2010). That is due to the fact that we are not including any processes that break this linearity such as thermal effects, pollutant deposition or turbulence induced by traffic.

Please also note the supplement to this comment:
http://www.atmos-chem-phys-discuss.net/acp-2016-202/acp-2016-202-AC2-
supplement.pdf

[Figure]

[Figure]

Figure 1. Vertical profiles of the spatial average of $k/u_\tau^2$ using 0.5 m and 1 m of grid resolution in the 2D geometry.

**Fig. 1.**

[Figure]

Figure 2. Vertical profiles of the spatial average of $U/u_\tau$ (left) and $k/u_\tau{}^2$ (right) using the fine and the coarse grid.

**Fig. 2.**

[Figure]

**Supplement:**

[revised manuscript text omitted]